Efficient processing of complex XSD using Hive and Spark

Martinez-Mosquera Diana diana.martinez@epn.edu.ec 1
Navarrete Rosa 1
Luján-Mora Sergio 2
1 Department of Informatics and Computer Science, Escuela Politecnica Nacional , Quito , Ecuador
2 Department of Software and Computing Systems, University of Alicante , Alicante , Spain
Shang Yilun
Electronic publication date: 2021 Aug 17
Publication date: 2021
Volume: 7
Electronic Location ID: e652
Received 2021 Apr 23; Accepted 2021 Jul 6
Copyright: ©2021 Martinez-Mosquera et al.
Copyright year: 2021
Copyright holder: Martinez-Mosquera et al.
License: This is an open access article distributed under the terms of the Creative Commons Attribution License, which permits unrestricted use, distribution, reproduction and adaptation in any medium and for any purpose provided that it is properly attributed. For attribution, the original author(s), title, publication source (PeerJ Computer Science) and either DOI or URL of the article must be cited.
License URL: https://creativecommons.org/licenses/by/4.0/

Keywords: Hive, Spark, Performance management, Mobile network, Complex XSD, XML

Funding: Unidad de Gestión de Investigación y Proyección Social from the Escuela Politécnica Nacional This work was supported by the Unidad de Gestión de Investigación y Proyección Social from the Escuela Politécnica Nacional. The funders had no role in study design, data collection and analysis, decision to publish, or preparation of the manuscript.

==============================
The eXtensible Markup Language (XML) files are widely used by the industry due to their flexibility in representing numerous kinds of data. Multiple applications such as financial records, social networks, and mobile networks use complex XML schemas with nested types, contents, and/or extension bases on existing complex elements or large real-world files. A great number of these files are generated each day and this has influenced the development of Big Data tools for their parsing and reporting, such as Apache Hive and Apache Spark. For these reasons, multiple studies have proposed new techniques and evaluated the processing of XML files with Big Data systems. However, a more usual approach in such works involves the simplest XML schemas, even though, real data sets are composed of complex schemas. Therefore, to shed light on complex XML schema processing for real-life applications with Big Data tools, we present an approach that combines three techniques. This comprises three main methods for parsing XML files: cataloging, deserialization, and positional explode. For cataloging, the elements of the XML schema are mapped into root, arrays, structures, values, and attributes. Based on these elements, the deserialization and positional explode are straightforwardly implemented. To demonstrate the validity of our proposal, we develop a case study by implementing a test environment to illustrate the methods using real data sets provided from performance management of two mobile network vendors. Our main results state the validity of the proposed method for different versions of Apache Hive and Apache Spark, obtain the query execution times for Apache Hive internal and external tables and Apache Spark data frames, and compare the query performance in Apache Hive with that of Apache Spark. Another contribution made is a case study in which a novel solution is proposed for data analysis in the performance management systems of mobile networks.

Introduction

The eXtensible Markup Language (XML) is now widely used on the Internet for different purposes. There are numerous XML-based applications that utilize tag-based and nested data structures (Chituc, 2017; Debreceny & Gray, 2001; Hong & Song, 2007) due to greater flexibility in the representation of different types of data: these can be customized by the user. However, the main constraint is that XML representation is inefficient in terms of processing and with respect to query times; for this reason, agile and intelligent search and query proposals are required (Hsu, Liao & Shih, 2012). XML streams represent an extremely popular form of data exchange used for social networks, Really Simple Syndication (RSS) feeds, financial records, and configuration files (Mozafari, Zeng & Zaniolo, 2012). It is therefore important to focus on efficiently processing data for complex semi-structured data such as XML.

The large amounts of XML data created daily have influenced the development of Big Data solutions to handle massive XML data in a scalable and efficient environment (Boussaid et al., 2006; Fan et al., 2018). Nowadays, a multitude of published studies have proposed methodologies and solutions for processing these types of files using Big Data tools. However, a more common approach in such works involves the simplest examples of XML documents, with few attributes or with fragments of the complete XML file, even though real data sets are composed of complex schemas that include nested arrays and structures. As explained in the Related Work section, the proposals generally test simple XML schemas or do not present the schema they are used for. Therefore, there is no proof they are useful in the real world. Moreover, it is complicated to reproduce the proposals as no detailed procedure is presented.

To address the lack of methods for processing XML files with complex schemas, we present an approach in this study based on three main methods: (1) cataloging, (2) deserialization, and (3) positional explode. In (1), we identify the main elements within an XML Schema Definition (XSD) and map them in a complete list of items with a systematic order: root, arrays, structures, values, and attributes. In (2), the XML file is converted into a table with rows and columns. Finally, in (3), the elements of the arrays are placed in multiples rows to improve the visualization for the final user.

To demonstrate the validity of our proposal in a Big Data environment, we present a case study that uses 3G and 4G performance management files from two mobile network vendors as real data sets. This is because 3G and 4G are now the most commonly used mobile technologies in the world (Jabagi, Park & Kietzmann, 2020). Furthermore, the mobile networks are growing at a rapid pace: according to the Global System for Mobile Communications, there were around 8 billion connections worldwide in the year 2020 and this is expected to reach 8.8 billion connections by 2025 (GSM, 2020). To provide mobile services to these users, thousands of network elements have been deployed around the world. These constantly generate performance management data to monitor the network status close to real-time. For this reason, this large amount of data must be queried in the shortest possible time to offer an excellent service to the end user and efficiently prevent or detect outages (Martinez-Mosquera, Navarrete & Lujn-Mora, 2020). These data sets are XML files composed of complex schemas with nested structures and arrays.

In this paper, we utilize an Apache Hadoop framework for the experiments as this is an open source solution that has been widely deployed in several projects. Moreover, it provides a distributed file system with the ability to process Big Data with both efficiency and scalability (Lin, Wang & Wu, 2013). In addition, our research addresses the evaluation of query execution times for complex XML schemas in different versions, with the aim of validating the proposal for old and new software developments.

For the test environment, the Hadoop Distributed File System (HDFS) was used as a data lake, as this is a powerful system that stores several types of data (Apache, 2021a). Additionally, we selected Apache Apache Hive due to its native support of XML files (Apache, 2021b), and the existing parsing serializer/deserializer tool from IBM to create the external and internal tables. Finally, we evaluate the execution of query times in Apache Spark (Apache, 2021c) through the implementation of data frames from XML files with complex schemas.

Using the proposed methods, tables and data frames can be created in a more intuitive form. We present all the processes involved in creating Apache Hive internal and external tables and Apache Spark data frames; the results of the evaluation of query execution times; and, finally, a comparison of the results between Apache Hive and Apache Spark.

Our main research questions are as follows:

1. Using the proposed method, is it possible to automatically create Apache Hive external and internal tables and Apache Spark data frames for complex schemas of XML files?

2. In terms of query execution time, what type of Apache Hive table is more efficient; internal or external?

3. Which system, Apache Hive or Apache Spark, provides the shortest query response times?

The remainder of this paper is organized as follows. In the next section, Related Concepts, we review concepts related to our work to facilitate understanding of this area. This section then presents existing relevant studies and the main contributions of our research. The Methods section presents the methods proposed to identify the elements of the catalog and the process to apply deserialization and positional explode methods in Apache Hive and Apache Spark. The Case Study section presents the experimental results of the query execution times for Apache Hive internal and external tables and Apache Spark data frames using 3G and 4G performance management files from two mobile network vendors. Finally, the Conclusions section draws final conclusions, answers the research questions, and discusses possible future work.

Related Concepts

To facilitate understanding, the following are brief descriptions of the main concepts employed in this work.

XML

XML (W3C, 2016) is a flexible text format that was developed by the XML working group of the World Wide Web Consortium (W3C) in 1996. It is based on the Standard Generalized Markup Language (SGML) or ISO 8879. The XML language describes a class of data objects called XML documents that are designed to carry data with a focus on what data are and not how data look. XML has been widely adopted as the language has no predefined tags. Thus, the author defines both the tags and the document structure. XML stores data in plain text format, making it human-readable and machine-readable. This provides an independent way of storing, transporting, and sharing data.

An XML schema describes the structure of an XML document and is also referred to as XSD. Within an XSD, the elements of the XML document are defined. Elements can be simple or complex. A simple element can contain only text, but they can be of several different types, such as boolean, string, decimal, integer, date, time, and so on. By contrast, a complex element contains other simple or complex types (W3C, 2016).

Complex XSD is considered in several studies (Krishnamurthy et al., 2004; Murthy & Banerjee, 2003; Rahm, Do & Mamann, 2004): those that use complex types, complex contents and/or extension bases on existing complex elements, or large real-world XSD. The complex contents specify that the new type will have more than one element. Extension bases refer to the creation of new data types that extend the structure defined by other data types (W3C, 2016). Large XSD is based on the number of namespaces, elements, and types in the XML files. Appendix A presents an example of a complex XSD that follows the 3rd Generation Partnership Project (2005) format used for performance management in mobile networks. In this work, the files used for the case study are based on this XSD.

The XSD can also define attributes that contain data related to a specific element. As best practice, attributes are used to store metadata of the elements and the data itself are stored as the value of the elements (W3C, 2016). The XML syntax below presents an example of the use of attributes and values in the elements: an attribute is used to store the identification number of the element and a value is used to store the data.

 <root>     <element attribute_id ="01">     value_of_element_01     </element>     <element attribute_id ="02">     value_of_element_02     </element>     </root>

Apache Hive

Apache Hive (Apache, 2021b) is a data warehouse software that incorporates its own query language based on SQL, named Apache HiveQL, to read and write data sets that reside in a distributed storage such as HDFS. Apache Hive also makes use of a Java Database Connectivity (JDBC) driver to allow queries from clients such as the Apache Hive command line interface, Beeline, and Hue. Apache Hive fundamentally works with two different types of tables: internal (managed) and external.

With the use of internal tables, Apache Hive assumes that the data and its properties are owned by Apache Hive and can only be changed via Apache Hive command; however, the data reside in a normal file system (Francke, 2021).

Conversely, Apache Hive external tables are created using external storage of the data,; for instance, the HDFS directory where the XML files are stored. The external tables are created in Apache Hive but the data are kept in HDFS. Thus, when the external table is dropped, only the schema in the database is dropped, not the data (Francke, 2021).

Apache Spark

Apache Spark (Apache, 2021c) is an engine for large data processing. It can work with a set of libraries such as SQL, Data Frames, MLlib for machine learning, GraphX, and Apache Spark Streaming. These libraries can be combined in the same application. Apache Spark can be used from Scala, Python, R, and SQL shells. This tool also runs on HDFS.

An Apache Spark data frame is a data set organized into named columns, similar to a table in a relational database. Data frames can be constructed from structured data files, tables in Apache Hive, external databases, or existing Resilient Distributed Data sets (RDD). An RDD is a collection of elements partitioned across the nodes of the cluster that can be operated in parallel. In general, RDD is created by starting with a file in the Hadoop file system (Apache, 2021c).

Performance management files in mobile networks

The mobile network sector is one of the fastest-growing industries around the world. Everybody has witnessed the rapid evolution of its technologies over the last few decades (Martinez-Mosquera, Navarrete & Lujn-Mora, 2020).

A mobile network is composed of Network Elements (NEs) that produce correlated Performance Measurement (PM) data managed by a Network Manager (NM) (3rd Generation Partnership Project, 2005). PM data are used to monitor the operator’s network, generate alarms in case of failures, and support decision-making in the area of planning and optimization. In summary, PM data check the behavior of network traffic, almost in real time, through the values of the measurements transmitted in every file. PM files are based on the XSD proposed in the Technical Specification 32.401 version 5.5.0 from 3rd Generation Partnership Project, 2005 which is presented in Appendix A. Each vendor then adapts and personalizes them according to their needs.

Related Work

A review of the literature identified research related to querying XML documents that involves numerous methods and algorithms. Most of these publications test their approaches using simple XSD, only a few are related to Apache Hive and Apache Spark. In the following section, we review the research most closely related and relevant to our study and explain how these studies differ from our approach. This comparison highlights some of the contributions of our work.

Hricov et al. (2017) evaluate the computation times to query XML documents stored in a distributed system using Apache Spark SQL and XML Path (XPath) query language to test three different data sets. The XML files contain only four attributes. They perform SQL queries to evaluate XPath queries and the Apache Spark SQL API. The main difference from our study is that we evaluate the query execution times for XML documents of complex types from real-life mobile networks in Apache Hive and Apache Spark. Furthermore, we explain in detail how to apply our proposed method to facilitate its replication in other studies, whereas Hricov et al. (2017) only summarize their proposal.

Luo, Liu & Watfa (2014) propose a schema to store XML documents in Apache Hive named open Apache Hive schema. The proposal consists of defining three columns: markup, content, and Uniform Resource Identifier (URI). Every tag is stored in the markup column, content refers to the value of the attribute and URI the data location. In contrast to our research, the cited study does not present examples for XSD with complex types, neither the results of the computation times in Apache Hive and Apache Spark. Moreover, the cited study only presents an approach for external tables, whereas our study includes internal tables and data frames for Apache Spark.

Hsu, Liao & Shih (2012) propose a system based on Apache Hadoop cloud computing framework for indexing and querying a large number of XML documents. They test the result times for streaming and batched query. They stated that the XML files need to be parsed and then indexes are produced to be stored as HDFS files; however, no details about the method employed to process the data is described, nor the XSD used in the research. They present the execution times obtained for the index construction and query evaluation. However, we explain in detail how to implement our approach using real data sets from mobile networks for Apache Hive and Apache Spark.

Hong & Song (2007) propose a method for permanently storing XML files into a relational database MS SQL Server. For tests, they employ a web-based virtual collaboration tool called VCEI. For each session, an XML format file is generated with four main entities: identification, opinion, location in the image, and related symbols. In this file, the opinions of the users are associated with digital images. A single table is created with the generated XML document. In contrast to our work, the authors do not focus on Big Data tools such as Apache Hive or Apache Spark, and they only use an XSD with simple types.

Madhavrao & Moosakhanian (2018) propose a method for combining weather services to provide digital air traffic data in standardized formats including XML and Network Common Data Form (NetCDF) using a Big Data framework. This work presents an example with an XML document. The reporting tool used is Apache Spark SQL, but no details about its implementation are presented. This approach focuses on providing a query interface for flight and weather data integration, but unlike our study does not evaluate query time execution in Apache Hive and Apache Spark.

Zhang & Mahadevan (2019) present a deep learning-based model to predict the trajectory of an ongoing flight using massive raw flight tracking messages in XML format. They cite the need to parse the raw flight XML files using the package ’com.databricks.Apache Spark.xml’ in Apache Spark to extract attributes such as arrival airport, departure airport, timestamp, flight ID, position, altitude, velocity, target position, and so on. However, no detail about the implementation is provided nor is there any information on the XSD used and the behavior for Apache Hive and Apache Spark that we consider in our study.

Vasilenko & Kurapati (2015) discuss the use of complex XML in the enterprise and the constraints in Big Data processing with these types of files. They thus propose a detailed procedure to design XML schemas. They state that the Apache Hive XML serializer-deserializer and explode techniques are suitable for dealing with complex XML and present an example of the creation of a table with a fragment of a complex XML file. By contrast, in our work, we propose the cataloging procedure and also evaluate our approach for Apache Spark data frames. We additionally present the query execution times obtained after applying our proposal.

Other studies also utilize XML documents to evaluate the processing time with HDFS and the Apache Spark engine; however, the files used for the tests contain simple types, with a few attributes, or do not present schemas  (Hai, Quix & Zhou, 2018; Kunfang, 2016; Zhang & Lu, 2021).

Finally, our research explains how to use cataloging, deserialization and positional explode to process complex XSD in Apache Hive internal and external tables and Apache Spark data frames; moreover, we demonstrate the validity of our proposal in a test Big Data environment with real PM XML files from two mobile network vendors.

Methods

The solution we propose for querying complex XML schemas is based on Big Data systems such as HDFS for storing, and Apache Hive and Apache Spark for reporting. Figure 1 sketches the architecture employed to evaluate the query execution times for Apache Hive and Apache Spark. HDFS provides a unified data repository to store raw XML files and external tables. In the reporting layer, Apache Hive and Apache Spark are connected to HDFS to perform the queries through Apache Hive Query Language (HQL) (Cook, 2018), and XPath expressions for Apache Hive (Tevosya, 2011) and Scala shell for Apache Spark (Apache, 2021c).

Figure 1 Hadoop architecture to evaluate the query execution times of complex XSD in Apache Hive and Apache Spark.

In general, Big Data architectures use the Extract Load and Transform (ELT) process (Marín-Ortega, Abilov & Gmez, 2014), which transforms the data into a compatible form, at the end of the process. The ELT process differs from the Extract Transform and Load(ETL) process, used for traditional data warehouse operations, where the transformation of the data is conducted immediately after the extraction (Mukherjee & Kar, 2017).

In the extract phase presented in Fig. 1, it is assumed that a cluster-based approach assigns a system to manage the collection of the XML files from the sources. Then, in the load phase, these data are stored in a repository such as HDFS. No changes in the format of the XML files are made in the load process; therefore the transformation process is not comprised.

Once the data are available on the HDFS, we identify three main vectors: (1) vector A = {Ai}; i = 1, …, N where N relates to the total number of XML files, (2) vector X = {Xi}; i = 1, …, M where M relates to the set of distinct XSD, and (3) vector E = {Ei}; i = 1, …, P where P relates to the different element types in the XSD according to the catalog defined in Table 1.

Table 1 Catalog to identify and depict element types from complex XSD.

Element types	Notation	
Root	< >	
Array	[]	
Structure	{}	
Attribute	@	
Value	#	

The catalog in Table 1 allows us to map the main element types present in an XML file with a possible notation through symbols such as < >, [], {}, @, and #. Our main goal is to facilitate the acknowledgment of XML element types to create tables and perform queries in both Apache Hive and Apache Spark. These symbols conform to the JavaScript Object Notation (JSON) (W3C, 2020) to avoid new syntax. Root type corresponds to the main tag used to initiate and terminate the XML file. Array and Structure types relate to the types of elements or children-elements array and structure, respectively. Attribute refers to the attributes and Value denotes the values of elements that can be of different types, such as strings, char, short, int, float, among others. For clarity, we present an example of the identification of vectors and its respective catalog for the XSD in Fig. 2.

Figure 2 Example of an XSD with complex types and extension base.

Figure 2 presents an example of a complex XSD. This contains the following complex types: element1, element11, element111, element112, and element12. Furthermore, element11 extends element111, and element111 extends element112.

For Fig. 2, there is an XML file only, thus vector A is identified with one element A1:

A = {A1}.

A1 is composed of one schema; thus vector X is composed by one element X1:

X = {X1}

Inside the X1 schema, the element types from Table 1 must be identified; therefore, vector E is composed of five vectors: (1) root E <  >, (2) arrays E[], (3) structures E{}, (4) attributes E@, and (5) values E#.

E = {E <  > , E[], E{}, E@, E#};

For Fig. 2, the five vectors contain the following elements:

1. E <  >  = {root}; //The main tag is the root.

2. E[] = {element1, element11}; //Two structures with elements of different data types.

3. E{} = {element111, element112, element12}; //Three arrays with elements of homogeneous data types.

4. E@ = {attribute111, attribute112, attribute12}; //Three attributes.

5. E# = {values_of_element111, values_of_element112, values_of_element12}; //Three elements with simple content.

After the XML element types are mapped with the proposed catalog in Table 1, tables for Apache Hive or data frames for Apache Spark can be created. Figure 3 summarizes the workflow for the three main methods: (1) cataloging, (2) deserialization, and (3) positional explode. For (1), the E <  > vector identifies the root of the used XSD, while E[] and E{} vectors allow easy identification of the indexes. An index is identified for each array. It is also important to state whether a structure is placed before an array; this structure also has an index. Without an index, internal queries to arrays load all the rows belonging to the array. But, with an index, only the specific record in a table is loaded.

Figure 3 Workflow to create tables and data frames for complex XSD.

Finally, the XPath queries to the nodes are determined for each element of the E@ and E# vectors. The indexes mapped also supply the expressions of XPath queries. The number of queries corresponds to the number of the columns in an Apache Hive column-separated table or Apache Spark data frame. The detailed procedure to create Apache Hive internal and external tables, and also Apache Spark data frames, is explained in the following sections.

Creation of Apache Hive tables for complex XSD

First, the input is the XML file denoted by Ai stored into HDFS. The XSD that belongs to Ai is named vector Xi. Vector E is composed of the elements root, array, structure, attribute, and value elements, identified after performing the map with the catalog proposed in Table 1.

_________________________________________________________________________________   Algorithm 1: Creation of Apache Hive Tables for Complex XSD          ____     Input:  XML documents Ai      Output:  Apache Hive Table  1  Create Xi  ← Ai;  2  Create E ← catalog;  3  while X  do      4   if  internal table  then     5   Create RawTable; 6   for Xi  do 7   xmlinput.start=<>" ← E <>; 8   xmlinput.end=< / >" ← E <> ; 9   location=/hdfs; 10   Deserialization ; 11   load data into table; 12   end 13   return RawTable; 14   Create ColumnSeparatedInternalTable; 15   for RawTable do 16   XPATH strings ← E@, E#; 17   Positional Explode ← E[], E{}; 18   return ColumnSeparatedInternalTable; 19   end 20   else     21   Create ExternalRawTable; 22   for Xi  do 23   xmlinput.start=<>" ← E <>; 24   xmlinput.end=< / >" ← E <>; 25   location=/hdfs; 26   Deserialization ; 27   end 28   return ExternalRawTable; 29   Create ColumnSeparatedExternalTable; 30   for ExternalRawTable do 31   XPATH strings ← E@, E#; 32   Positional Explode ← E[], E{}; 33   return ColumnSeparatedExternalTable; 34   end 35   end 36  end

Apache Spark data frames for complex XSD

The first phase in the creation of the Apache Hive table relies on XmlInputFormat from the Apache Mahout project (Holmes, 2014) to shred the input file into XML fragments based on specific start and end tags, determined in the vector E <  >. The XML Deserializer queries the XML fragments with an XPath Processor to populate column-separated Apache Hive tables, where the number of columns correspond to the element numbers of the vector E@ plus E#. However, method (3) may be needed to explode the array data into multiple rows with the help of the elements identified in the vectors E[] and E{}.

The entire process is summarized in Algorithm 1. The input is the XML file denoted by Ai, in which the XML Schema and the vector E are identified. Create RawTable or Create ExternalRawTable are applied using the deserialization method, the output of which serves as the input to the Create ColumnSeparatedInternalTable or Create ColumnSeparatedExternalTable functions, respectively. In these functions, positional explode methods are applied. Once the workflow is completed, the data from the XML files are stored in rows and columns in Apache Hive tables and queries through HQL can be performed. This process is useful for internal and external tables.

As stated in the Related Concepts section, Apache Spark can work with a set of libraries. In this work, we select Data Frames as it is similar to the column-separated Apache Hive tables and is independent of other database engines such as Apache Hive.

According to the methods proposed in Fig. 3, the use of the catalog, deserialization, and explode methods are also suitable for the Apache Spark engine. The difference in Apache Spark, is that there are no internal and external table concepts; therefore, the XML is stored in a data frame variable and the queries are performed over these data frames. Similar to Apache Hive, arrays must be exploded, and values and attributes are the fields to be queried.

The entire process for creating data frames in Apache Spark is summarized in Algorithm 2. The input is the XML file denoted by Ai stored previously in HDFS, and the XSD that belongs to Ai is named vector Xi. Vector E is composed of the root, array, structure, attribute, and value elements, identified after performing the map with the catalog proposed in Table 1.

The function CreateDataFrame is applied to vector Xi using the deserialization method to return the data frame. The output of CreateDataFrame is the input of the CreateColumnSeparatedDataFrame function, where the number of elements in vectors E@ and E# correspond to the number of columns and elements in vectors E[] and E{} for the positional explode method. Finally, the data frame with column-separated values is returned.

__________________________________________________________________________________________   Algorithm 2: Creation of Apache Spark Data Frames for Complex XSD____     Input:  XML documents Ai      Output:  Apache Spark Data Frame  1  Create Xi  ← Ai;  2  Create E ← catalog;  3  while X  do      4   CreateDataFrame; 5   for Xi  do     6   rowTag=<>" ← E <>; 7   location=/hdfs; 8   Deserialization ; 9   end 10   return DataFrame; 11   Create ColumnSeparatedDataFrame; 12   for DataFrame do     13   Select Expression ← E@, E#; 14   Positional Explode ← E[], E{}; 15   return ColumnSeparatedDataFrame; 16   end 17  end

Appendices B and C present examples of the Apache Hive tables and Apache Spark data frames created for the example file in Appendix A.

Case Study

The implementation of the Big Data framework was performed using a cloud computing solution composed of a virtual machine VM.Standard 2.2 with the following hardware features:

• 200 GB of storage.

• Two VCPU cores and two threads per core with 4 GHz in total.

• 30 GB of RAM.

At software level, we tested our proposal in two environments. It was important to evaluate two versions to verify that the proposal is applicable in any version and that the query times improve in recent versions. Furthermore, not all potential users always have access to the latest versions of the software.

Version 1. Apache Hadoop HDFS version 2.6.0, Apache Hive version 1.1.0, Apache Spark version 1.6.0, Java version 1.7.0_67 and Scala version 2.10.5.

Version 3. Apache Hadoop HDFS version 3.2.1, Apache Hive version 3.1.2, Apache Spark version 3.0.1, Java version 1.8.0_271, and Scala version 2.12.10.

For clarity, we present a case study using PM XML files taken from two mobile network vendors. As stated in the Introduction section, these data are selected because mobile networks generate a high volume of these files every second. For instance, in the United States of America in 2019, there were 395,562 cell sites (Statista, 2020). Therefore, taking a PM file with 20 KB as reference, the total file size is analyzed to be approximately 8 GB per second. These samples are based on the XSD presented in Appendix A according to the 3GPP standard. The 3GPP standard maps the tags defined in the file format definition to those used in the XML file (3rd Generation Partnership Project, 2005). The data sets used for the tests are the following:

1. An PM XML file from a real 3G mobile network from a vendor named A with the schema presented below:     mdc     |-- md: array     |     |-- element: struct     |     |     |-- mi: struct     |     |     |     |-- gp: long     |     |     |     |-- mt: array     |     |     |     |     |-- element: string     |     |     |     |-- mts: string     |     |     |     |-- mv: array     |     |     |     |     |-- element: struct     |     |     |     |     |     |-- moid: string     |     |     |     |     |     |-- r: array     |     |     |     |     |     |     |-- element: string     |     |     |-- neid: struct     |     |     |     |-- nedn: string     |     |     |     |-- nesw: string     |     |     |     |-- neun: string     |-- mff: struct     |     |-- ts: string     |-- mfh: struct     |     |-- cbt: string     |     |-- ffv: string     |     |-- sn: string     |     |-- st: string     |     |-- vn: string

2. An PM XML file from a real 4G mobile network from a vendor named B with the schema presented below:    measCollecFile     |-- fileFooter: struct     |     |-- measCollec: struct     |     |     |-- #VALUE: string     |     |     |-- __endTime: string     |-- fileHeader: struct     |     |-- __fileFormatVersion: string     |     |-- __vendorName: string     |     |-- fileSender: struct     |     |     |-- #VALUE: string     |     |     |-- __elementType: string     |     |-- measCollec: struct     |     |     |-- #VALUE: string     |     |     |-- __beginTime: string     |-- measData: struct     |     |-- managedElement: struct     |     |     |-- #VALUE: string     |     |     |-- __userLabel: string     |     |-- measInfo: array     |     |     |-- element: struct     |     |     |     |-- __measInfoId: long     |     |     |     |-- granPeriod: struct     |     |     |     |     |-- #VALUE: string     |     |     |     |     |-- __duration: string     |     |     |     |     |-- __endTime: string     |     |     |     |-- measTypes: string     |     |     |     |-- measValue: array     |     |     |     |     |-- element: struct     |     |     |     |     |     |-- __measObjLdn: string     |     |     |     |     |     |-- measResults: string     |     |     |     |     |     |-- suspect: boolean     |     |     |     |-- repPeriod: struct     |     |     |     |     |-- #VALUE: string     |     |     |     |     |-- __duration: string

In this research, we evaluate the query execution times for Apache Hive and Apache Spark after applying the workflow proposed in Fig. 3. As described in the Method section, we utilized the XSD of two PM XML files from mobile network vendors named A and B. Table 2 presents the catalog mapping of the XSD elements for the A and B mobile network vendors. The Root column identifies the roots of the used XSD, Arrays identifies all the tags with arrays, Structures identifies all the tags with structures, and Attributes and Values presents all the attributes and values of the XML files.

Once the arrays in the XSD are identified in Table 2, this column allows easy identification of the indexes. Thus, an index is identified for each array; however, as mentioned previously, it is important to observe whether a structure is placed before an array; this structure will also have an index. For instance, Table 3 presents the indexes identification for the XSD from mobile network vendors A and B. For the second array identified in Table 3 for vendor A, mt array belongs to the mi structure; therefore, mt.index and mi.index are mapped. Similarly, for the first array identified in Table 3 for vendor B, measInfo array belongs to the measData structure; therefore, measInfo.index and measData.index are mapped.

Table 2 Cataloging and deserialization for XSD from mobile network vendors A and B.

Vendor	Root	Arrays	Structures	Attributes and values	
A	mdc < >	md[]	mff{}	mi{}gp#	
		md[]mi{}mt[]	mfh{}	mi{}mts#	
		md[]mi{}mv[]	md[]mi{}	element{}moid#	
		md[]mi{}mv[]r[]	mv[]element{}	r[]element#	
			md[]neid{}	neid{}nedn#	
				neid{}nesw#	
				neid{}neun#	
				mff{}ts#	
				mfh{}cbt#	
				mfh{}ffv#	
				mfh{}sn#	
				mfh{}st#	
				mfh{}vn#	
				mt[]element@	
B	measCollecFile < >	measData{}measInfo[]	fileFooter{}	fileHeader{}_fileFormatVersion#	
		measInfo[]measValue[]	fileFooter{}measCollec{}	fileHeader{}_vendorName#	
			fileHeader{}	measInfo[]measTypes#	
			fileHeader{}fileSender{}	measInfo[]_measInfoID#	
			fileHeader{}measCollec{}	measValue[]_measObjLdn#	
			measData{}	measValue[]measResults#	
			measData{}managedElement{}	measValue[]suspect#	
			measInfo[]grandPeriod{}	measCollec{}_endTime@	
			measInfo[]repPeriod{}	measCollec{}_beginTime@	
				fileSender{}_elementType@	
				measData{}_userLabel@	
				grandPeriod{}_duration@	
				grandPeriod{}_endTime@	
				repPeriod{}_duration@	

Table 3 Identification of indexes for XSD from mobile network vendors A and B.

Vendor	Array	Index	
A	md[]	md.index	
	md[]mi{}mt[]	mi.index	
		mt.index	
	md[]mi{}mv[]	mv.index	
	md[]mi{}mv[]r[]	r.index	
B	measData{}measInfo[]	measData.index	
		measInfo.index	
	measInfo[]measValue[]	measValue.index	

Finally, the XPath queries are determined for each attribute and value identified in Table 2. The indexes mapped in Table 3 also supply the expressions of the XPath queries. For instance, in the first row of Table 4, the XPath expression should be mdc/md/mi/gp; however, to query only the gp, the XPath expression uses the md.index and mi.index indexes.

Table 4 Positional explode and query identification for XSD from mobile network vendors A and B.

Vendor	Attributes and values	XPath	Query	
A	mi{}gp#	/mdc/md.index/mi.index/gp	Q1	
	mi{}mts#	/mdc/ md.index/mi.index/mts	Q2	
	element{}moid#	/mdc/md.index/mi.index/mv.index/moid	Q3	
	r[]element#	/mdc/md.index/mi.index/mv.index/r.index/element	Q4	
	neid{}nedn#	/mdc/md.index/neid/nedn	Q5	
	neid{}nesw#	/mdc/md.index/neid/nesw	Q6	
	neid{}neun#	/mdc/md.index/neid/neun	Q7	
	mff{}ts#	/mdc/mff/ts	Q8	
	mfh{}cbt#	/mdc/mfh/cbt	Q9	
	mfh{}ffv#	/mdc/mfh/ffv	Q10	
	mfh{}sn#	/mdc/mfh/sn	Q11	
	mfh{}st#	/mdc/mfh/st	Q12	
	mfh{}vn#	/mdc/mfh/vn	Q13	
	mt[]element@	/mdc/md.index/mi.index/mt.index/element	Q14	
B	fileHeader{}_fileFormatVersion#	/measCollecFile/fileHeader/_fileFormatVersion	Q1	
	fileHeader{}_vendorName#	/measCollecFile/fileHeader/_vendorName	Q2	
	measInfo[]measTypes#	/measCollecFile/measData.index/measInfo.index/measTypes	Q3	
	measInfo[]_measInfoID#	/measCollecFile/measData.index/measInfo.index/_measInfoId	Q4	
	measValue[]_measObjLdn#	/measCollecFile/measData.index/measInfo.index/measValue.index/ _measObjLdn	Q5	
	measValue[]measResults#	/measCollecFile/measData.index/measInfo.index/measValue.index/ measResults	Q6	
	measValue[]suspect#	/measCollecFile/measData.index/measInfo.index/measValue.index/suspect	Q7	
	fileFooter{}_endTime	/measCollecFile/fileFooter/measCollec/_endTime	Q8	
	measCollec{}_beginTime	/measCollecFile/fileHeader/measCollec/_beginTime	Q9	
	fileSender{}_elementType	/measCollecFile/fileHeader/fileSender_elementType	Q10	
	measData{}_userLabel	/measCollecFile/measData.index/managedElement/_userLabel	Q11	
	grandPeriod{}_duration	/measCollecFile/measData.index/measInfo.index/granPeriod/_duration	Q12	
	grandPeriod{}_endTime	/measCollecFile/measData.index/measInfo.index/granPeriod/_endTime	Q13	
	repPeriod{}_duration	/measCollecFile/measData.index/measInfo.index/repPeriod/_duration	Q14	

The number of queries corresponds to the number of the columns in a column-separated table or data frame. Table 4 presents the identification of 14 queries for mobile network vendors A and B, where the column-separated table or data frame will contain 14 columns for each vendor. It is a simple coincidence that they both have 14 queries as this can vary from file to file. The results obtained are presented in the following subsections.

Results

Apache Hive

XML files with 4,668 rows and 8,461 rows were used for the tests of mobile network vendors A and B, respectively. Preliminary experiments were conducted to determine the average query execution time for a complete table. Versions 1 and 3 were tested in four different scenarios:

1. Creating an Apache Hive external raw table and then an Apache Hive view table with the positional explode method.

2. Creating an Apache Hive internal raw table and then an Apache Hive view table with the positional explode method.

3. Creating an Apache Hive external raw table and then a new Apache Hive table with the positional explode method.

4. Creating an Apache Hive internal raw table and then a new Apache Hive table with the positional explode method.

It is important to highlight that it is necessary to create the raw table to deal only with parent labels of the XML file as we are parsing complex schemas (Intel, 2013), and, furthermore, positional explode is only available for SELECT sentences (Microsoft, 2021; Databricks, 2021). For these preliminary experiments, we utilized the Data Query Language (DQL) type from HQL for query statements and no limits in the rows were expressed.

Table 5 presents the average query execution times in the four scenarios and for the two versions of Big Data software components. As indicated, the first and second scenarios take longer than the third and fourth scenarios. Figure 4 presents the data for Table 5 in graphical form, where the difference is remarkable. Therefore, based on the results of these preliminary tests, the first and second scenarios were discarded.

Table 5 Average query execution times (seconds) for XML files from vendors A and B in fourth scenarios.

Scenario	Vendor A [s] version 1	Vendor B [s] version 1	Vendor A [s] version 3	Vendor B [s] version 3	
1	185.04	344.43	0.26	0.25	
2	184.84	341.64	0.23	0.26	
3	0.19	0.30	0.13	0.13	
4	0.18	0.67	0.11	0.10	

Figure 4 Average query execution times (seconds) for XML files from vendors A and B in fourth scenarios.

Following the base expression of Appendix B, we created Apache Hive external and internal raw tables and then column-separated tables with the positional explode method. Over these tables, we conducted the queries obtained for each mobile network vendor from Table 4 using sentences of DQL type with HQL. We limited the query to 1,000 rows in order to perform the tests in a common scenario for all queries and tools.

In this section, we present the results of the evaluation of execution query time for:

1. Apache Hive internal tables, where the XML files and the Apache Hive tables are stored in the same Apache Hive directory and the queries are performed through HQL.

2. Apache Hive external tables, where the XML files are stored in HDFS and the tables are stored in a Apache Hive directory. Queries are also performed through HQL.

Figure 5 presents the average query execution times for the XML files from vendor A and Fig. 6 from vendor B. The queries relate to those identified in Table 4 for the third scenario in blue for Apache Hive version 1 and yellow for version 3, and the fourth scenario in red for version 1 and purple for version 3. The x axis presents the query identifications and the y axis the average query execution time in ms . To determine the optimal query execution time, the sample mean, variance, and standard deviation are calculated.

Figure 5 Average query execution times (milisenconds) for apache hive external and internal tables versions 1 and 3 from vendor A.

Figure 6 Average query execution times (milisenconds) for Apache Hive external and internal tables versions 1 and 3 from vendor B.

To calculate the sample mean X ¯, we denote the observations drawn from the Apache Hive external and internal tables by XEi and XIi respectively, with i = 1, …, 14, and N = 14 according to the number of queries. Let: X ¯E=1N∑i=1NXEiandX ¯I=1N∑i=1NXIi

To calculate the variance σ ˆ2, let the expression: σ ˆE2=1N−1∑i=1NXEi−X ¯E2andσ ˆI2=1N−1∑i=1NXIi−X ¯I2

To obtain the standard deviation σ ˆ, the square root of the variance is calculated.

As Table 6 indicates, for external tables, the query execution time for vendor A deviates from the average by approximately 22.53 ms for Apache Hive version 1 and 8.37 ms for Apache Hive version 3. For internal tables, the standard deviation is equal to 12.75 ms for Apache Hive version 1 and 2.67 ms for Apache Hive version 3.

Table 6 Mean, variance, and standard deviation for query execution times (milliseconds) in Apache Hive external and internal tables.

Vendor	Hive	Mean XE ¯	Mean XI ¯	Variance σ ˆE2	Variance σ ˆI2	Standard deviation σ ˆE	Standard deviation σ ˆI	
A	1	118.21	90.71	507.72	162.53	22.53	12.75	
	3	100.50	77.07	70.12	7.15	8.37	2.67	
B	1	101.07	82.57	246.07	130.42	15.69	11.42	
	3	82.43	75.50	41.80	6.88	6.47	2.62	

Conversely, for vendor B, the query execution time deviates from the average by approximately 15.69 ms for Apache Hive version 1 and 6.47 for Apache Hive version 3 for external tables. The standard deviation for internal tables is 11.42 ms for Apache Hive 1 and 2.62 ms for Apache Hive 3. Therefore, the standard deviation obtained from Apache Hive external tables for versions 1 and 3 is greater than that for internal tables; thus, we conclude the fourth scenario allows lower query execution time and that the query execution time for Apache Hive version 3 is more efficient than version 1 as we expected.

We perform other tests in order to determine the behavior for external and internal Apache Hive tables version 3, with different file sizes. We only test version 3 as this provides a better performance. The results are presented in Fig. 7.

Figure 7 Average query execution times (ms ) for internal and external Apache Hive tables with different XML file sizes.

As explained in the Related Concepts section, an internal Apache Hive table stores data in its own directory in HDFS, while an external Apache Hive table uses data outside the Apache Hive directory in HDFS. Therefore, as expected, the query execution times for internal tables are smaller than external tables. However, as indicated in Fig. 7, as the number of rows in an XML file increases, internal Apache Hive tables perform better than external tables. For instance, for 3,000,000 rows the query execution time takes approximately 400 ms , while for an external table it takes around 1,800 ms .

Apache Spark

We also evaluated the query execution times for XSD from mobile network vendors A and B in the Apache Spark engine versions 1 and 3. First, a data frame with a single row of raw data was created as positional explode is only available for SELECT sentences. Like Apache Hive, XML files with 4668 rows and 8461 rows were used for the tests of mobile network vendors A and B, respectively.

The tests were conducted for the same queries employed for Apache Hive from Table 4. Again, we limited the query to 1,000 rows. Each query was performed in the Scala shell and follows the query syntax of Data Retrieval Statements (DRS). For instance, to query Q1 of the mobile network vendor A:

   var dataframe = sqlContext.read.    format("com.databricks.Apache  Spark.xml").    option("rowTag", "mdc").load("/hdfs")    dataframe.selectExpr("explode(md)as_md").    select(\$"_md.mi.gp").show(1000)

In this section, we present the results of the evaluation of query execution times for Apache Spark data frames. XML files are also stored in HDFS. Queries are performed through a domain-specific language for structured data manipulation in the Scala shell.

The attained results for the query execution times in ms for Apache Spark are presented in Table 7. Because there are no internal and external table concepts; only one execution time is obtained for each query. As Table 7 indicates, for vendor A the query execution time deviates from the average by approximately 323.19 ms for Apache Spark version 1 and 60.15 ms for version 3. Conversely, for vendor B the standard deviation is equal to 287.31 ms for version 1 and 71.86 ms for version 3. An important conclusion based on these results is that our proposal can be applied on different versions of Apache Spark and performance in the more recent versions is improved.

Table 7 Query execution times, mean and standard deviation (milliseconds) in Apache Spark version 1 and 3.

Vendor /	
Apache Spark	
Version	Q1	Q2	Q3	Q4	Q5	Q6	Q7	Q8	Q9	Q10	Q11	Q12	Q13	Q14	X ¯	σ ˆ	
A/1	1850	1840	1770	1360	1880	1030	1130	2050	1390	1850	1380	1300	1260	1600	1549.29	323.19	
A/3	193	217	307	219	143	158	118	193	166	86	106	103	108	177	163.86	60.15	
B/1	1880	1550	1980	1770	1800	1240	1100	1840	1920	1340	1480	1560	1290	1850	1614.29	287.31	
B/3	243	168	423	253	239	186	249	187	133	139	206	216	157	204	214.50	71.86	

Comparison between Apache Hive and Apache Spark

According to the results of the case study, Apache Hive and Apache Spark are useful for processing complex XML schemas using our proposed method. Figure 8 presents a comparison of query execution times between the Apache Hive external table and the Apache Spark data frame for a single row of raw data. Version 3 is used for the reasons stated previously. Furthermore, we use the Apache Hive external table because the raw data is inside the HDFS and the queries are performed there directly. From these results, we conclude that the external Apache Hive table is more efficient when queries to a complete data frame are performed, as indicated in Fig. 8. Appendices B and C present examples of the sentences used to create the raw table for Apache Hive and the data frame with a single row in Apache Spark, utilizing the XSD from Appendix A.

Figure 8 Average query execution times (ms ) comparison for Apache Hive and Apache Spark with a single row of raw data.

Additionally, Figs. 9 and 10 compares the query execution times between Apache Spark data frames and Apache Hive external and internal tables for the 14 queries identified in Table 4. The results indicate that, for every individual query, the attained times for Apache Spark are 100 times or more greater than those attained for Apache Hive. Therefore, query execution times for Apache Hive are also lower for individual values or attributes than Apache Spark.

Figure 9 Average query execution times (miliseconds) for Apache Hive and Apache Spark from vendor A.

Figure 10 Average query execution times (miliseconds) for Apache Hive and Apache Spark from vendor B.

Similar results for individual queries can be observed in Table 6, where the query execution time for Apache Hive version 1 deviates from the average by 22.53 ms as a maximum value for vendor A. Conversely, as indicated in Table 7, the query execution time for Apache Spark version 3 deviates from the average by 60.15 ms as a minimum value.

We can therefore conclude that, because Apache Hive is only a database engine for data warehousing, where the data are already stored in tables inside HDFS as its default repository, it exhibits better performance than Apache Spark, Apache (2021b). Moreover, Apache Spark is not a database even though it can access external distributed data sets from data stores such as HDFS. Apache Spark is able to perform in-memory analytics for large volumes of data in the RDD format; for this reason, an extra process is needed over the data, Apache (2021c). For queries over XML files with complex schemas, Apache Spark is no more efficient than Apache Hive; however, Apache Spark works better for complex data analytics in terms of memory and data streaming, Ivanov & Beer (2016).

Comparison with other approaches

As mentioned previously, multiple studies have evaluated the processing of XML files with Big Data systems. However, these approaches involve the simplest XML schemas and are generally not suitable for complex schemas that are more common in real life implementations.

The studies by Hai, Quix & Zhou (2018); Hricov et al. (2017) and Hsu, Liao & Shih (2012) present the results of their experiments processing XML files in terms of query execution time. However, the features of their Big Data ecosystem differ from ours and they do not present the versions used for the software. Therefore, we only take as reference the query execution time of Hricov et al. (2017) that is approximately 7 s for 1,000,000 rows; while, as a result of our work, to query approximately 3,000,000 rows, Apache Hive external tables take around 2 s; while for the Apache Spark data frame the queries take around 14.25 s, using the Big Data environment version 3.

Conclusions and Future Work

Motivated by the need to evaluate queries for complex XSD that are now used in multiple applications, and the Big Data solutions available for processing this file format, we proposed three main methods to facilitate the creation of Apache Hive internal and external tables, Apache Spark data frames, and the identification of their respective queries based on the values and attributes of the XML file.

The three proposed methods were (1) cataloging, (2) deserialization, and (3) positional explode. In (1), five element types of an XSD were identified: root, arrays, structures, values, and attributes. The root element identification facilitated the creation of a raw table with the content of the XML file. In (2), identification of attributes and values elements allowed the raw XML data to be converted into a table with rows and columns. Finally, in (3), the arrays were placed in multiples rows to improve the visualization to the final user.

To validate our proposal, we implemented a Big Data framework with two versions of software components named version 1 and 3. As a case study, we used the performance management files of 3G and 4G technologies from two mobile network vendors as real data sets. Using the proposed methodology, internal and external Apache Hive tables and Apache Spark data frames were created in a more intuitive form for both versions. Finally, we presented the execution times of the 14 identified queries for PM files from mobile network vendors A and B. The query types used in this work can be employed for other data sets as they are composed only of SELECT statements.

The experimental results indicated that query execution times for Apache Hive internal tables performed better than Apache Hive external tables and Apache Spark data frames. Moreover, the Big Data environment implemented with HDFS version 3.2.1, Apache Hive version 3.1.2, Apache Spark version 3.0.1, Java version 1.8.0271, and Scala version 2.12.10 exhibited better performance than with older versions.

Another important point to make is that the results of direct queries to a data frame with a single row took longer than queries to a Apache Hive external table. Based on these results, our research questions are answered as follows:

1. It is possible to create Apache Hive external and internal tables and Apache Spark data frames using our proposed method. For the cataloging process, the following elements are identified: root, structures, arrays, attributes, and values. For the deserialization process, the values and attributes are populated into column-separated fields, while for the positional explode the arrays are uncompounded into multiple rows.

2. Apache Hive internal tables generate lower query execution times than external tables with the fourth proposed scenario: creating an Apache Hive internal raw table and then creating a new Apache Hive table with the positional explode method. This result is consistent with the expected behavior as tables and data are stored in the same directory in HDFS.

3. When comparisons are made between Apache Hive and Apache Spark, Apache Hive external table allows for shorter query execution times when queries to a complete data frame are performed. Moreover, Apache Hive external or internal tables are more efficient than Apache Spark for queries to individual values or attributes. We believe this occurs because Apache Spark requires extra in-memory processing for queries on XML files.

In future work, we plan to explore the behavior of a Big Data cluster with several nodes. Moreover, we plan to include PM files from 5G mobile networks in the tests, and to create a benchmark for different data sets and queries.

Supplemental Information

Supplemental Information 1 Performance management XML file from mobile network vendor A

The file contains the raw data from a real 3G mobile network vendor. The XML schema is based on the 3GPP TS 32.401 standard.

Click here for additional data file.

Supplemental Information 2 Performance management XML file from mobile network vendor B

The file contains the raw data from a real 3G mobile network vendor. The XML schema is based on the 3GPP TS 32.401 standard.

Click here for additional data file.

Supplemental Information 3 Results of the query execution times, means, variances and standard deviation for Hive and Spark version 1

There are two tabs. Hive tab presents the results of the query execution times, means, variances, and standard deviation for Hive internal and external tables from 14 queries identified in the proposed methodology. Spark tab presents the results of the query execution times, means, variances, and standard deviation for Spark data frames from 14 queries identified in the proposed methodology.

Click here for additional data file.

Supplemental Information 4 Results of the query execution times, means, variances and standard deviation for Hive and Spark version 3

There are three tabs. Hive tab presents the results of the query execution times, means, variances, and standard deviation for Hive internal and external tables from 14 queries identified in the proposed methodology. Spark tab presents the results of the query execution times, means, variances, and standard deviation for Spark data frames from 14 queries identified in the proposed methodology. Different XML file sizes tab presents the results of the query execution times for different file sizes.

Click here for additional data file.

Supplemental Information 5 Measurement collection data file XML schema according to 3GPP TS 32.401

The file extracts the measurement collection data file XML schema for performance management files in mobile networks in A.4.2 section, and the mapping of ASN.1 Measurement Report File Format tags to XML tags Table A.1, from the 3GG TS 32.401 standard.

Click here for additional data file.

APPENDIX A. 3GPP TS 32.401 V5.5.0 PM data file XSD

   <?xml  version="1.0" encoding="UTF-8"?>    <!--      3GPP TS 32.401 PM Concept and Requirements      Measurement collection data file XML schema      measCollec.xsd    -->    <schema       targetNamespace=    "http://www.3gpp.org/ftp/specs/latest/rel-5/32_series         /32401-540.zip#measCollec"       elementFormDefault="qualified"       xmlns="http://www.w3.org/2001/XMLSchema"       xmlns:mc=    "http://www.3gpp.org/ftp/specs/latest/rel-5/32_series         /32401-540.zip#measCollec"    >       <!-- Measurement collection data file root XML element           -->       <element  name="measCollecFile">         <complexType>            <sequence>              <element  name="fileHeader">                <complexType>                   <sequence>                     <element  name="fileSender">                        <complexType>                          <attribute  name="localDn"                          <attribute  name="elementType" type="                              string" use="optional"/>                        </complexType>                     </element>                     <element  name="measCollec">                        <complexType>                          <attribute  name="beginTime" type="                              dateTime" use="required"/>                        </complexType>                     </element>                   </sequence>                   <attribute  name="fileFormatVersion" type="                       string" use="required"/>                   <attribute  name="vendorName" type="string" use                       ="optional"/>                   <attribute  name="dnPrefix" type="string" use="                       optional"/>                </complexType>                        </element>                        <element name="measData" minOccurs="0" maxOccurs="                            unbounded">                           <complexType>                             <sequence>                               <element name="managedElement">                                 <complexType>                                   <attribute name="localDn" type="string"                                       use="optional"/>                                   <attribute name="userLabel" type="string                                       " use="optional"/>                                   <attribute name="swVersion" type="string                                       " use="optional"/>                                 </complexType>                               </element>                               <element name="measInfo" minOccurs="0"                                   maxOccurs="unbounded">                                 <complexType>                                   <sequence>                                     <element name="granPeriod">                                        <complexType>                                          <attribute                                            name="duration"                                            type="duration"                                            use="required"                                          />                                          <attribute                                            name="endTime"                                            type="dateTime"                                            use="required"                                          />                                        </complexType>                                     </element>                                     <choice>                                        <element name="measTypes">                                          <simpleType>                                            <list itemType="Name"/>                                          </simpleType>                                        </element>                                        <element name="measType"                                                 minOccurs="0" maxOccurs="                                                     unbounded">                                          <complexType>                                            <simpleContent>                                              <extension base="Name">                                                <attribute name="p"                                                            type="                                                                positiveInteger                                                                " use="                                                                required"/>                                              </extension>                                            </simpleContent>                                          </complexType>                                        </element>                                     </choice>                                     <element name="measValue"                                               minOccurs="0" maxOccurs="                                                   unbounded">                                        <complexType>                                          <sequence>                                            <choice>                                              <element name="measResults">                                                <simpleType>                                                  <list itemType="                                                      mc:measResultType"/>                                                </simpleType>                                              </element>                                              <element name="r"                                                        minOccurs="0"                                                            maxOccurs="                                                            unbounded">                                                <complexType>                                                  <simpleContent>                                                     <extension base="                                                        mc:measResultType">                                                       <attribute name="p"                                                           type="                                                           positiveInteger"                                                                                     use                                                                                         =                                                                                         "                                                                                         required                                                                                         "                                                                                         /                                                                                         >                                                     </extension>                                                  </simpleContent>                                                </complexType>                                              </element>                                            </choice>                                            <element name="suspect" type="                                                boolean" minOccurs="0"/>                                          </sequence>                                          <attribute name="measObjLdn"                                                      type="string" use="                                                         required"/>                                        </complexType>                                     </element>                                   </sequence>                                 </complexType>                               </element>                             </sequence>                           </complexType>                        </element>                        <element name="fileFooter">                           <complexType>                             <sequence>                               <element name="measCollec">                                 <complexType>                                   <attribute name="endTime" type="dateTime                                       " use="required"/>                                 </complexType>                               </element>                             </sequence>                           </complexType>                        </element>                      </sequence>                    </complexType>                  </element>                  <simpleType name="measResultType">                    <union memberTypes="decimal">                      <simpleType>                        <restriction base="string">                           <enumeration value="NIL"/>                        </restriction>                      </simpleType>                    </union>                  </simpleType>                </schema>

APPENDIX B. Apache Hive Table Creation

   CREATE  TABLE RawTable ( xml STRING )    ROW FORMAT SERDE "com.ibm.spss.Apache  Hive.serde2.xml.         XmlSerDe"    WITH SERDEPROPERTIES ( "column.xpath.xml"="/" )    STORED    AS    INPUTFORMAT "com.ibm.spss.Apache  Hive.serde2.xml.         XmlInputFormat"    OUTPUTFORMAT "org.apache.hadoop.Apache  Hive.ql.io.         IgnoreKeyTextOutputFormat"    LOCATION "/hdfs/tmp"    TBLPROPERTIES    ("xmlinput.start"="<MeasDataCollect",     "xmlinput.end"="</MeasDataCollect>");    CREATE  TABLE ColumnSeparatedTable    AS    SELECT REGEXP_EXTRACT(t.INPUT__FILE__NAME, ’ˆ.*/(.*)$’, 1)                                                      XPATH_STRING    (t.xml,"/MeasDataCollect/MeasFileHeade/fileFormatVersion")    XPATH_STRING    (t.xml,CONCAT("/MeasDataCollect//MeasData[", MeasuData.         index, "]/NEID/NEUserName"))    FROM RawTable    AS t    LATERAL VIEW MeasData    AS  index    WHERE MeasData.index  > 0;

APPENDIX C. Apache Spark Data Frame Creation

   var dfApache Spark=sqlContext.read.format("com.databricks.         Apache  Spark.xml").    option("rowTag","MeasDataCollect").    load("/hdfs/tmp").    dfApache Spark.selectExpr("explode(MeasData)as_md").    select($"_md.neid.neun").    show()

Additional Information and Declarations

Competing Interests

Author Contributions

Data Availability

The authors declare there are no competing interests.

Diana Martinez-Mosquera conceived and designed the experiments, performed the experiments, analyzed the data, performed the computation work, prepared figures and/or tables, and approved the final draft.

Rosa Navarrete and Sergio Luján-Mora conceived and designed the experiments, performed the experiments, analyzed the data, prepared figures and/or tables, authored or reviewed drafts of the paper, and approved the final draft.

The following information was supplied regarding data availability:

The raw measurements, results of the experiments, and part of the referenced standard 3GPP are available in the Supplemental Files.

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
