# Peer review of "Efficient processing of complex XSD using Hive and Spark"

_PeerJ Computer Science, doi:10.7717/peerj-cs.652_

## Round 0.1 · original submission · Major Revisions

We have received two reviewers for the paper. Both reviewers found some merits of the paper but they also pointed out some drawbacks. The experiment design and methodology need to be further clarified. Please provide detailed responses to the reviewers. Note that you should not cite references recommended by reviewers if not appropriate.

Reviewer 1 ·

Basic reporting

The authors have a good structure for the paper, however, there are grammatical errors and sentence construction errors in multiple places. I have described a few examples below

"However, a more common approach in that works involves the simplest examples of XML documents, even though, the real data sets are composed of complex schemas that include nested arrays and structures."
"The reporting tool used is Spark SQL but no details about the implementation are presented."
"With the purpose of reducing the lack of methods for processing XML files with complex schemas, in this study we present our approach based on three main methods: (1) catalog, (2) deserialization, and (3) positional explode."

I would recommend them to correct these

Experimental design

The problem statement that the authors are trying to solve is not clearly stated.
Are they trying to compare big data frameworks on fast they can process complex XSD or are they trying to prove that their approach based on cataloging, deserialization and positional explode is superior to other approaches in the related work?
if they are trying to do the former, they need to consider several performance characteristics. Big data frameworks usually run on clusters, not on a single nodes.
Apache spark version used (1.6.0) is outdated and retired, I would recommend the authors to use Apache Spark 3.0 since its performance is almost that in 1.6.0
The same goes for Apache Hive. For end customers, the reason to do internal tables and external tables are very different, however, that choice does affect performance. The authors have not disclosed whether while using internal tables, their results are affected by caching in Hive.

Validity of the findings

While the authors prove that their approach of catalog, deserialization and positional explode works in Big Data Frameworks, they have not compared that with other approaches for XML parsing that the related works have described

The authors have not explored why Hive or Spark performs better. When they mention Hive performs better for queries to extract individual values or attributes, what is the reason behind this? They need to explore the open-source code to understand what is the root cause. This would improve the validity of their findings.

Additional comments

Please compare with the latest version of the big data frameworks since they are more up-to date and the scan processing time is shorter there.
Please explore deeper into the frameworks to find the reasons for your findings
Also explore big data cluster results

·

Basic reporting

I appreciate authors for their research contribution. The paper is well-organized and contributes to novel research work which falls in Computer Science Research domain of the journal.

1. Few sentences which need to be re framed/reexamined as some how these sentences meaning is not clear. Line numbers are mentioned below:
46-48
193-194
273-274
280-281
307-308

2. I found some of the fundamental papers related to work done. Authors can check and include these in related work or wherever it seems suitable:

Dmitry Vasilenko, “An Empirical Study on XML Schema Idiosyncrasies in Big Data Processing”, in International Journal on Computer Science and Engineering, October 2015.
Dmitry Vasilenko, Mahesh Kurapati,.” Efficient Processing of XML Documents in Hadoop Map Reduce, IJCSE, 2014, Vol.6, No.9,p.329–333.
Song Kunfang and Hongwei Lu, “Efficient Querying Distributed Big-XML Data using MapReduce”, Int. J. Grid High Perform. Comput. 8, 3 (July 2016), 70–79. DOI:https://doi.org/10.4018/IJGHPC.2016070105

Experimental design

1. At line number 444-445- I request if you can elaborate or reference why it is needed to create the raw table at first?

2. Please discuss which type of queries have been selected to evaluate the proposed algorithm. Also, mention whether the same type of queries can be applicable for other application datasets?

Validity of the findings

No Comment

Additional comments

1. The paper is devoted to important task of Big Data. The authors have presented query processing algorithms for complex XML files. The practical value of article is good.

2. The proposed algorithms can be further tested on different size big datasets to validate them for implementation on real big datasets.

3. For further research, authors can take benchmark datasets and queries.

All the best.

---

## Round 0.2 · accepted · Accept

The paper can be accepted. Congratulations.

·

Basic reporting

I appreciate authors for working on all observations. The proof reading of document further ensure clear and unambiguous English. Hence, the re submission is acceptable for publishing purpose.

Experimental design

Since, authors have answered all observations in detail, I am satisfied with the improved version of document. No more questions from my side.

Validity of the findings

All findings are validated properly. No further queries.

Additional comments

All previously given observations are considered and corrected in re-submission by authors. .All the best for future endeavors.